# Validation of Inertial-Measurement-Unit-Based Ex Vivo Knee Kinematics during a Loaded Squat before and after Reference-Frame-Orientation Optimisation

**DOI:** 10.3390/s24113324

**Published:** 2024-05-23

**Authors:** Svenja Sagasser, Adrian Sauer, Christoph Thorwächter, Jana G. Weber, Allan Maas, Matthias Woiczinski, Thomas M. Grupp, Ariana Ortigas-Vásquez

**Affiliations:** 1Research and Development, Aesculap AG, 78532 Tuttlingen, Germanyallan.maas@aesculap.de (A.M.); thomas.grupp@aesculap.de (T.M.G.); ariana.ortigas_vasquez@aesculap.de (A.O.-V.); 2Department of Orthopaedic and Trauma Surgery, Musculoskeletal University Center Munich (MUM), Campus Grosshadern, Ludwig Maximilians University Munich, 81377 Munich, Germany; christoph.thorwaechter@med.uni-muenchen.de (C.T.); m.woiczinski@waldkliniken-eisenberg.de (M.W.); 3Experimental Orthopaedics University Hospital Jena, Campus Eisenberg, Waldkliniken Eisenberg, 07607 Eisenberg, Germany

**Keywords:** kinematics, optimisation, knee joint, local reference frame, IMU, knee rig

## Abstract

Recently, inertial measurement units have been gaining popularity as a potential alternative to optical motion capture systems in the analysis of joint kinematics. In a previous study, the accuracy of knee joint angles calculated from inertial data and an extended Kalman filter and smoother algorithm was tested using ground truth data originating from a joint simulator guided by fluoroscopy-based signals. Although high levels of accuracy were achieved, the experimental setup leveraged multiple iterations of the same movement pattern and an absence of soft tissue artefacts. Here, the algorithm is tested against an optical marker-based system in a more challenging setting, with single iterations of a loaded squat cycle simulated on seven cadaveric specimens on a force-controlled knee rig. Prior to the optimisation of local coordinate systems using the REference FRame Alignment MEthod (REFRAME) to account for the effect of differences in local reference frame orientation, root-mean-square errors between the kinematic signals of the inertial and optical systems were as high as 3.8° ± 3.5° for flexion/extension, 20.4° ± 10.0° for abduction/adduction and 8.6° ± 5.7° for external/internal rotation. After REFRAME implementation, however, average root-mean-square errors decreased to 0.9° ± 0.4° and to 1.5° ± 0.7° for abduction/adduction and for external/internal rotation, respectively, with a slight increase to 4.2° ± 3.6° for flexion/extension. While these results demonstrate promising potential in the approach’s ability to estimate knee joint angles during a single loaded squat cycle, they highlight the limiting effects that a reduced number of iterations and the lack of a reliable consistent reference pose inflicts on the sensor fusion algorithm’s performance. They similarly stress the importance of adapting underlying assumptions and correctly tuning filter parameters to ensure satisfactory performance. More importantly, our findings emphasise the notable impact that properly aligning reference-frame orientations before comparing joint kinematics can have on results and the conclusions derived from them.

## 1. Introduction

In the field of orthopaedics, the use of joint kinematic data to objectively quantify patient function and mobility before and after treatment can markedly improve medical outcomes [1,2,3]. Current gold standard technologies are often considered to be static [4,5] or dynamic [3,6,7] fluoroscopy or in some cases even marker-based optical motion capture systems [8,9]. Both fluoroscopic and optical systems are unfortunately not only time- and cost-intensive but also require a large laboratory space and the involvement of experienced technicians [9,10], thereby limiting their use in regular clinical workflows. In recent years, inertial measurement units (IMUs) have been explored as a cheaper and more flexible alternative to the aforementioned gold standards [11]. At the most basic level, IMUs consist of at least (1) a gyroscope that measures angular velocity and (2) an accelerometer that measures linear acceleration. The accuracy and reliability of the kinematic signals estimated from the measured inertial datapoints highly depend on the performance and robustness of the sensor fusion algorithm implemented, as well as the ability to address sources of errors, like drift [9,12]. Consequently, the thorough validation of said algorithms is a crucial step towards the standard application of IMU-based gait analysis systems in a clinical setting, which could in turn lead to an improvement in orthopaedic patient care and satisfaction.

Previously, a particular implementation [13] of a Rauch–Tung–Striebel smoother was selected for further exploration based on a number of advantages: flexible sensor placement, non-susceptibility to ferromagnetic disturbances, and no need for extensive calibration [14]. A first set of tests to explore the algorithm’s accuracy in the absence of soft tissue artefacts was designed, in which real knee motion patterns that had been previously collected in vivo during level walking, stair decent and sit-to-stand using moving videofluoroscopy [15] were used to guide a six degrees of freedom joint simulator. Knee kinematics estimated based on the two IMUs that had been rigidly attached to the simulator demonstrated promising accuracy for level walking, but errors were larger for the other activities, especially the more they differed from a standard gait. Notably, every simulator trial consisted of at least 50 iterations of each activity cycle. On one hand, this number of repetitions was necessary to ensure accurate execution of the motion by the robotic simulator; on the other hand, however, it could have represented an unrealistic advantage for the performance of the sensor fusion algorithm (especially considering the recursive nature of Rauch–Tung–Striebel smoothing, where a forward pass based on an extended Kalman filter is followed by a backward recursion smoother, and accuracy generally improves with additional iterations) [16].

In this study, in order to further evaluate the effectiveness and reliability of a promising IMU-based gait analysis system [13,14,17], a prototype was tested in a different, more challenging experimental setup. This served as an intermediate step between the initial tests that relied on the controlled execution of simulated data and future in vivo tests in real-use cases. Originally designed and conducted as part of a separate study, experiments consisted of simulating a single loaded squat in each of seven cadaveric specimens using a force-controlled knee rig. By additionally attaching IMU sensors to that knee rig setup, IMU-based rotational knee kinematics could be estimated and compared against an optical-marker-based reference system. Moreover, the potential effects of inconsistencies between the local reference frames defined by the IMU- vs. optical-based motion capture systems were subsequently assessed by using the REference FRame Alignment MEthod (REFRAME) [17,18]. Differences in the kinematic signals stemming from each system were then evaluated both before and after the implementation of REFRAME. In this manner, this study represents an additional step in the validation process of a promising IMU-based gait analysis system, bringing us closer to its possible application in clinics.

## 2. Materials and Methods

### 2.1. Experimental Setup

Data used for this investigation were collected as part of a study that was approved by the ethics committee of the Ludwig Maximilians University Munich (ID 58-16). Data collection and management complied with all the relevant institutional, national, and international guidelines and legislation. Seven cadaveric knees (fresh frozen specimens; 3 female; 2 right; aged 80.4 ± 4.6 years; Table 1) were tested on the Munich knee rig, an established force-controlled device [19,20,21]. Exclusion criteria included any previous surgical intervention to the specimen’s knee or hip, as well as any records of symptoms pointing to a musculoskeletal pathology of the knee. Moreover, legs with a varus/valgus deformity greater than 10° were excluded from the study. Notably, qualitative observations on bone quality and soft tissue conditions were recorded and are available upon specific request. Kinematic profiles were collected strictly to assess agreement between measuring systems and should not necessarily be interpreted as representative of a “healthy” knee joint. Three additional specimens that were originally tested unfortunately had to be excluded from the analyses because of missing datapoints (e.g., due to corrupted or missing raw data files from either the optical or IMU systems). Specimen tests were carried out by five of the listed study authors (see Author Contributions; Investigation), including the laboratory supervisor, the lead knee rig operator, and the developer of the IMU system prototype used. Specimen handling was additionally overseen by an experienced orthopaedic surgeon. All specimen tests took place within a span of four months. Each specimen was tested within 36 h of thawing, as well as within 60 h of having been removed from a freezing temperature.

Bone cuts were made 20 cm proximally and 22 cm distally from the epicondylar line, after which the femur and tibia were embedded in metal pots with epoxy resin (RenCast FC 52/53 Isocyanate & FC 53 Polyol, Huntsman Advanced Materials GmbH, The Woodlands, TX, USA). A cortical screw was used to attach the fibula head to the proximal tibia, and a constant muscle force of 20 N was applied throughout the entire load cycle by attaching metallic finger traps (Bühler-Instrumente Medizintechnik GmbH, Tuttlingen, Germany) to the vastus medialis, vastus lateralis, musculus semitendinosus, and biceps femoris. The mentioned loads were applied in the directions of the respective muscle origins to simulate a physiological line of action (Figure 1; for more information on the direction of forces see Figure 2 in [19]). Further details on this general setup have been described in prior studies [21,22,23].

A squat from approximately 30° to 130° of the knee flexion (as measured by two angular sensors placed on the hip and ankle joint, respectively; 8820, Burster, Gernsbach, Germany) was performed with a constant angular velocity of 3°/s. In line with previous studies that demonstrated that the shapes of kinematic profiles do not change considerably by increasing ground reaction forces, a controlled muscle force was applied to the rectus femoris to achieve a constant ground reaction force of 50 N. As per Müller et al. [24], further increasing the load would unnecessarily stress both the specimens and equipment but not guarantee better qualitative outcomes. Rotational tibio-femoral kinematics were calculated based on the measurements from two different systems. One system consisted of two IMUs (Xsens Dot, Movella, Enschede, Netherlands), where the sensors were fixed to the metal pots at the ends of the femur and tibia using custom 3D printed parts (Figure 1, right, in blue). Furthermore, the second system comprised a high-resolution 3D camera (Aramis, GOM, Braunschweig, Germany) and 2D optical markers attached to 3D printed parts fixed to the femur, tibia and patella (Figure 1, left, in black).

### 2.2. Calculation of Tibio-Femoral Kinematics

IMUs were used to sample linear acceleration and angular velocity at 60 Hz. Estimates of tibio-femoral joint angles were obtained from these raw inertial measurements based on a sensor fusion algorithm that leveraged Rauch Tung Striebel smoothing (i.e., extended Kalman filtering and smoothing) as per Versteyhe et al. [12,13,14,25]. While an earlier formulation of the algorithm included an offset correction step that relied on the assumption that the subject would start and end the trial at a neutral reference pose with 0° of knee flexion, this operation was adapted to account for a starting flexion angle of approximately 30° by calibrating the flexion value at the first timepoint to match that of the marker-based system.

A high-resolution camera system was used to track the 3D coordinates of selected anatomical landmarks based on rigid clusters of adhesive reflective markers (Figure 1). A right-handed global coordinate system was automatically determined by the camera system software such that the general directions of the x-, y- and z- axes were up, left and towards the camera setup, respectively. Local reference frames were defined for both the femur and tibia segments. The medio-lateral axis of the femoral reference frame was oriented laterally, in the direction of a vector connecting the medial and lateral epicondyles, with the femoral origin located at the midpoint between the two landmarks. The antero-posterior axis of the femur was defined to be positive anteriorly, orthogonal to the medio-lateral axis and a vector connecting the fossa intercondylaris to a point along the longitudinal axis of the femoral shaft, at the centre of the proximal surface of the top metal pot. Lastly, the femoral proximo-distal axis pointed proximally, in a direction orthogonal to the previously defined medio-lateral and antero-posterior axes. For the tibial reference frame, the origin was set at the midpoint between the tubercles of the tibial intercondylar eminence. The tibial medio-lateral axis pointed laterally in the direction of a vector pointing from the medial to the lateral tibial condyle. The tibial antero-posterior axis, on the other hand, pointed anteriorly, orthogonal to the medio-lateral axis and a vector from the centre of the tibial intercondylar eminence to a point along the longitudinal axis of the tibial shaft, at the centre of the distal surface of the bottom metal pot. The proximo-distal axis was then defined as orthogonal to the first two axes, pointing in the proximal direction.

For both IMU and optical systems, joint rotations were expressed as the orientation of the local tibia frame relative to the local reference frame of the femur. Moreover, Cardan angles were calculated based on an intrinsic XYZ rotation sequence from the femoral to the tibial frame, where the x-axis was pointed laterally, the y-axis anteriorly and the z-axis proximally for a right knee (left knees were mirrored into right knees). The knee joint angles in the sagittal, frontal and transversal planes corresponded to extension(+)/flexion(−), adduction(+)/abduction(−) and internal(+)/external(−) rotation, respectively. In this manner, two sets of kinematic signals were derived for each trial: one based on inertial and the other on optical measurements. These kinematic signals were then plotted over the progression of the activity cycle and the root-mean-square errors (RMSEs) between data capture systems were calculated for rotations around each of the three axes. RMSE was chosen as one of the most common metrics used to assess the performance of predictive models [26], like the one leveraged by the IMU-based algorithm, and effectively consists of calculating the square root of the average of the squared differences between observed (here, optical) and predicted (here, IMU) outcomes.

### 2.3. REFRAME

The primary flexion axis assumed by the optical system was defined based on the position of bony landmarks, as described above. On the other hand, the IMU-based system instead leveraged functional calibration methods (finding a best fit primary axis by first approximating the knee joint as a perfect hinge and then expanding the model to consider a ball-and-socket joint) to define the analogous joint axis. Given that natural knees do not behave as perfect hinges, the anatomical axis assumed by the optical system will inevitably differ to some degree from the functional axis used by the inertial system. Notably, previous studies have highlighted the importance of accounting for differences in local reference-frame orientations prior to comparing joint movement patterns based on kinematic signals [17]. Consequently, after the two aforementioned sets of rotational kinematics had been calculated (inertial and optical), REFRAME [17,18] was applied to each of the datasets, to optimise the orientation of the associated local segment reference frames and thus ensure consistency in our comparison of the joint angles.

Two different implementations of the REFRAME approach were explored. The first (REFRAME_IMU→GOM_; i.e., “optimising IMU towards GOM”) minimised the RMSEs of the IMU-based kinematic signals versus the optical-based estimates. Additionally, frame transformations consisting of rotations around the femoral x-axis during optimisation were restricted to prevent non-physiological frame orientations. Rotations around the different segment frame axes have an intuitive clinical meaning only within certain ranges (e.g., if the femoral longitudinal axis is roughly in the direction of the femoral shaft or the vector connecting the knee joint centre to the hip joint centre). Mathematically, however, it would be possible to, e.g., keep a consistent clinically meaningful flexion angle between femoral and tibial frames (by rotating both frames simultaneously), even while using a set of local frame orientations that no longer holds clinical meaning (e.g., if the longitudinal axes of the femur and tibia local segment reference frames are both at a 45° from the direction of the respective bone shafts). Restricting REFRAME transformations consisting of rotations around the mediolateral axis for one of the two segment frames is therefore a way to avoid this effect. The goal of this implementation was to determine whether, despite any initial apparent differences between the inertial and optical joint kinematic signals (due to differences in local reference-frame orientation), the underlying motion being characterised was in fact the same. Given the described experimental setup, both systems were known to measure the same underlying joint motion and, in the absence of errors, their kinematic signals should therefore coincide after the implementation of REFRAME. This method would inherently transform the orientation of the IMU-based reference frames to match that of the optical based frames, so optimisation of the inertial dataset was not entirely independent as it relied on information contained within the optical dataset.

A second implementation of REFRAME (REFRAME_RMS_) was also explored that independently minimised the root-mean-square (RMS) of abduction/adduction and external/internal rotation signals (both with criteria weighting of 1). Optimisation rotations around the femoral x-axis were once again restricted. Finally, in an effort to maintain clinical interpretability after REFRAME, flexion/extension was anchored to its raw values by minimising the RMSE between raw and optimised signals, with a criterion weighting of 0.1.

After REFRAME analysis, the resulting kinematic signals were once again plotted over the progression of the activity cycle, and the RMSEs between data-capture systems were calculated for rotations around each of the three axes. Paired t-tests at a 5% significance level were executed to compare RMSEs before and after REFRAME_IMU→GOM_, as well as before and after REFRAME_RMS_. The sensor fusion algorithm, joint angle estimates, REFRAME implementation, and RMSE calculations were all performed in MATLAB (vR2022b; The Mathworks Inc., Natick, MA, USA). Paired *t*-tests were performed in GraphPad Prism 10 (v10.1.0; GraphPad Software Inc.; San Diego, CA, USA).

## 3. Results

An analysis of the IMU- and GOM-based kinematics before REFRAME (“raw”) showed agreement only between flexion/extension signals (Figure 2). Clear differences were visible between the two systems for abduction/adduction and external/internal rotation, with RMSEs up to 20.4° ± 10.0° for abduction/adduction and 8.6° ± 5.7° for external/internal rotation (Table 2). Notably, the magnitude of errors affecting abduction/adduction angles was seemingly associated with the joint flexion, possibly indicating the presence of a crosstalk artefact.

Nevertheless, after the first REFRAME implementation, REFRAME_IMU→GOM_, there was a significant improvement in the agreement between the two datasets for all three movement directions (RMSE of 3.0° ± 2.0° for flexion/extension, 0.9° ± 0.4° for abduction/adduction and 1.4° ± 0.7° for external/internal rotation) (Figure 3, Table 2). REFRAME_IMU→GOM_ led to transformations of the local femoral reference frame of 0° around the x-axis (restricted by the optimisation formulation), 13.5° ± 13.7° around the y-axis, and 2.0° ± 3.1° around the z-axis, on average (Table 3). On the other hand, the tibia frame was on average transformed by 0.7° ± 4.5° around the x-axis, 19.3° ± 10.8° around the y-axis and 4.4° ± 6.9° around the z-axis. (Although traditional mean and standard deviation values are provided here for context, attempts at interpretation should consider that standard operators act non-commutatively when dealing with transformations.)

A second self-contained implementation of the REFRAME approach, REFRAME_RMS_, whereby the optimisation of a dataset was achieved using only information contained within itself, likewise resulted in visible improvement in the agreement of the inertial and optical kinematic signals for two of the joint angles (RMSE of 0.9° ± 0.4° for abduction/adduction and 1.5° ± 0.7° for external/internal rotation) (Figure 4, Table 2). Only for flexion/extension was there a slight deterioration in the correspondence of the IMU and GOM signals (RMSE of 4.2° ± 3.6°) (Table 2). On average, the changes in frame orientations associated with the IMU signals resulting from the implementation of REFRAME_RMS_ were 15.0° ± 10.1° and 0.6° ± 0.7° around the y- and z- femoral axes, respectively (Table 4). Changes to the orientation of the tibial frame, on the other hand, averaged 2.0° ± 2.1° around x, 13.6° ± 9.2° around y and 6.1° ± 3.7° around z. Analogously, the orientation of the GOM-based femoral reference frame was transformed by 1.4° ± 4.3° around y and −1.4° ± 3.0° around z, on average (Table 5). Finally, the GOM tibial frame was rotated 0.0° ± 0.6° around the x-axis, −5.9° ± 4.3° around the y-axis and 2.0° ± 3.4° around the z-axis.

The results of paired *t*-tests indicated that the change in RMSE for flexion/extension was not statistically significant for either REFRAME implementation (Figure 5a). However, the decreases in RMSE values for abduction/adduction and external/internal rotation brought about by the REFRAME application were found to be statistically significant for a *p*-value of 0.05 (Figure 5b,c). In fact, even after Bonferroni correction to account for the double comparison (raw vs. after REFRAME_IMU→OPT_, and raw vs. after REFRAME_RMS_), the decrease in the RMSEs of out-of-sagittal-plane rotations was still considered statistically significant.

## 4. Discussion

An IMU-based tool capable of accurately capturing tibio-femoral joint angles during activities of daily living could be extremely valuable in improving patient outcomes in orthopaedic care. In previous work, we adapted and performed first-level validation testing of a prototype system consisting of two IMUs and the implementation of an extended Kalman filter and smoother algorithm [14]. Testing occurred under idealised conditions using a six-degrees of freedom joint simulator and fluoroscopy-based data collected in vivo. Additional work then explored the implementation of a frame orientation optimisation method [17], demonstrating its relevance in ensuring valid conclusions were reached regarding the similarity of and/or difference in joint movement patterns. Importantly, this work determined that even minor inconsistencies in the alignment of joint axes can lead to unreliable kinematic signals. In this study, we therefore tested seven cadaver specimens on a force-controlled knee rig to evaluate the accuracy of the aforementioned IMU-based system in estimating joint kinematics under more challenging conditions than previously tested. This represented a valuable transition from testing on a robotic simulator to testing on real cadaveric specimens, before eventually progressing to the intended in vivo testing conditions. The simulated squat movement was additionally analysed using an optical marker-based system for reference. To ensure reliable comparisons, the kinematic signals stemming from both motion-capture systems were processed using REFRAME, in an effort to align the underlying local coordinate systems of the IMU- and optical-marker-based systems.

Prior to the implementation of REFRAME, our results revealed agreement between the IMU and GOM kinematics for flexion/extension only, showing clear differences for abduction/adduction and external/internal rotation. Moreover, standard deviation values revealed that the implemented IMU algorithm performs differently on different subjects (at least within our limited sample of seven), and these variations seemed to remain even after REFRAME implementation, an effect that was not clearly evident when assessing multiple iterations of repetitive motion patterns, like gait. Importantly, the Rauch Tung Striebel smoother assumed by default a reference pose of 0° flexion, 0° adduction and 0° tibial rotation. In the previous study using a six-degrees of freedom joint simulator [14], the assumption of a perfectly neutral reference pose was accurate. In the current study, however, specimens were known to start closer to 30° of flexion, and so the flexion angle at the reference pose assumed by the IMU-based system was appropriately adjusted. Reference abduction/adduction and external/internal rotation values were not analogously adapted, under the assumption that they would be negligible. The resulting discrepancies between the IMU and GOM signals for abduction/adduction and external/internal rotation observed at the beginning of most cycles therefore highlights the importance of having appropriate estimates for all three joint angles in the reference pose to obtain reliable kinematic measurements, especially in the absence of post-processing methods, like REFRAME.

The patterns of errors in abduction/adduction and external/internal rotation seen in the raw kinematic patterns are clearly indicative of cross-talk, which refers to how differences in the alignment of the knee’s medio-lateral axis lead to an artificial increase in the amplitude of out-of-sagittal plane rotations (Figure 2). In an idealised representation of cross-talk, with a frame misalignment of a non-zero angle around one of the three frame axes, what should be perceived as pure flexion around a single axis would instead result in artefact non-zero rotations around the remaining two axes (for a visual representation of this effect, please refer to Supplementary Figure S1 of Ortigas Vasquez et al. [17]). Around one axis, this effect would follow a sine wave pattern, starting at zero with zero flexion, peaking at 90° of flexion and progressively decreasing back to zero by 180° of flexion. This is the case for int/external rotation in Figure 2 (c.f. specimens 4, 6 and 7), as errors peak at approximately 90° of flexion, beyond which they decrease back down again until peak flexion halfway through the cycle, only for the effect to be mirrored as the knee extends in the second half of the squat. Around the other secondary axis, the artefact would behave like a cosine wave, starting at its peak high (or low) at 0° of flexion, reaching zero at 90° of flexion and continuing on to reach its peak low (or high) at 180° of flexion. This error pattern is clearly reflected for ab/adduction in Figure 2 (c.f. specimens 4, 6 and 7), where the error progressively increases, even past 90° of flexion, and peaks at peak flexion and then decreases back down with extension. Notably, for specimen 3 (Figure 2), the GOM-based kinematic signals for out-of-sagittal-plane rotations were more visibly indicative of crosstalk than the IMU-based signals, highlighting that neither GOM-based nor IMU-based kinematics are necessarily more objectively “accurate” than the other, but rather just graphically different.

The fact that the knee joint does not behave like a proper hinge (which can only rotate about a single fixed axis), but instead displays the complex motion pattern of a six-degrees-of-freedom joint, poses a unique challenge. For activities of daily living (e.g., level walking, stair descent, squat) any true instantaneous axis of rotation will in general not have a constant orientation relative to the bony anatomy, which fundamentally implies that any joint axis identified by palpating bony landmarks on, e.g., the distal femur, will be inherently “flawed” or at least different. This logic suggests that approximating a landmark-based joint axis using only IMU data (and thus functional methods) would be next to impossible. The first REFRAME implementation, REFRAME_IMU→GOM_, was therefore a necessary step to ascertain whether the differences observed between IMU- and GOM-based signals could potentially be explained by inconsistencies in local reference-frame orientations, rather than reflecting definitive differences in joint motion. After applying REFRAME_IMU→GOM_, there was a visible improvement in the agreement between the IMU and GOM systems for all three joint angles, suggesting that indeed most of the differences observed between the kinematic signals could be explained by differences in the orientations of the joint axes identified by the two systems. After REFRAME_IMU→GOM_ had been implemented to ensure a consistent axis orientation, there was considerable improvement in the agreement between IMU- and GOM-based signals for all three rotations, especially abduction/adduction and external/internal rotation (Figure 3). This suggests that REFRAME effectively reduced the impact of crosstalk artefacts, drastically improving the agreement to under 3° of deviation between joint angles around all axes. While certainly sufficient for most applications, given the small range of motion in, e.g., external/internal rotation, errors of 1.4° ± 0.7° (Table 2) may still be critical in some use cases (especially in combination with soft tissue artefacts), suggesting that further work to improve system accuracy could still be beneficial. Importantly, this analysis revealed that, while the IMU system was subject to some measurement error, most of the initially observed differences between IMU and GOM actually stemmed from problems with calibration of the sensor fusion algorithm, rather than IMU sensor inaccuracies.

The transformations applied during REFRAME_IMU→GOM_ (Table 3) effectively describe the rotations needed to align the IMU-based local femoral and tibial reference frames with those of the GOM system. These transformations can most likely be attributed to two key components. The first is the difference in frame orientations that results from the “incorrect” assumption of 0° abduction and 0° tibial rotation at the first timepoint. For example, according to the GOM system, specimen 3 begins the squat cycle with over 20° of abduction (Figure 2). As a result, reconciliation of the IMU and GOM signals will undeniably demand a change of approximately 20° in the relative orientation of the raw IMU femoral and tibial frames. This is substantiated by the results, as REFRAME_IMU→GOM_ leads to the transformation of the IMU femoral and tibial frames around the corresponding y-axes by −6.2° and 14.2°, respectively. The second key component contributing to the transformations implemented by REFRAME stems from fundamental differences in the types of joint axes identified by the two systems. While the GOM system defines joint axes based on the 3D coordinates of specific anatomical landmarks, IMU-based kinematics are restricted to purely functional methods. Angular deviations between functional and anatomical axes have been previously estimated to be 1°–5° in the knee joint [27,28,29]. Consequently, up to 5° (or potentially more considering crosstalk) of the frame transformations by REFRAME_IMU→GOM_ could easily be attributed to such differences.

The ideal post-processing approach we envision to ensure consistent frame orientations and positions fulfils three key criteria: (1) the resulting signals are clinically interpretable, (2) it should ensure consistent frame orientations and positions regardless of the initial choice of raw frames, and (3) it should be self-contained (i.e., aside from the objective criteria fed by the user, optimisation should rely exclusively on information contained within the set of signals being optimised itself). Although REFRAME_IMU→GOM_ fulfilled the first two, the third was violated; the implementation relied on the GOM-based signals, optimising the IMU-based signals towards them. Despite this shortcoming, REFRAME_IMU→GOM_ provided valuable insight into just how much of the differences between the raw IMU and GOM signals could potentially be explained by reference-frame alignment inconsistencies. RMSE values after REFRAME_IMU→GOM_ thus quantified the magnitude of the measurement error that could not possibly be attributed to frame alignment issues. In contrast, the second REFRAME implementation, REFRAME_RMS_, did fulfil the independence criterion; each dataset was optimised without any input from its counterpart. The minimisation of abduction/adduction and external/internal rotation RMS inherently minimised crosstalk artefacts in both sets of kinematic signals [17], leading to improved agreement between the systems (as evidenced by the reduction in RMSEs of out-of-sagittal-plane rotations compared to raw RMSEs). Notably, the objective criteria for REFRAME_RMS_ only slightly considered (criteria weighting of 0.1) changes to the flexion/extension, thereby resulting in a slight increase in this rotation’s RMSE. The fact that RMSEs after REFRAME_RMS_ were marginally larger than after REFRAME_IMU→GOM_ suggests that although consistency in frame alignment improved after both optimisations, the level of convergence between local reference frames achieved after REFRAME_RMS_ was not quite as precise as with REFRAME_IMU→GOM_. This helpfully illustrates the challenges associated with trying to fulfil all three key criteria with a single method, which in our experience is extremely difficult (if not impossible). Finally, in terms of the frame transformations applied to the femoral and tibial frames as part of REFRAME_RMS_ optimisation (Table 4 and Table 5), the combined magnitudes of individual rotations strongly support the assumption that almost the same level of convergence was reached as with REFRAME_IMU→GOM_. For example, REFRAME_IMU→GOM_ transformations indicated that for specimen 3, frame alignment required a −6.2° rotation around the femoral *y*-axis. Similarly, REFRAME_RMS_ led to femoral frame transformations of 3.0° around y for the IMU frames and 9.1° for the GOM frames, totalling about −6.1° net relative rotation around y, much like with REFRAME_IMU→GOM_. Analogous effects were observed in all other subjects.

In conclusion, our study demonstrates the importance of incorporating an approach, such as REFRAME, when evaluating the so-called accuracy (or rather, agreement) between different motion-capture technologies that are known to rely on different methods of joint axis definition. Not only are consistent reference-frame orientations crucial for a robust comparison of kinematic signals, REFRAME analysis has the potential to reveal valuable information about the possible sources of error affecting our measures. Importantly, our work also emphasises the difficulty in developing an ideal post-processing method for reference-frame alignment, especially in light of what are conflicting, sometimes even mutually exclusive, objectives. Finally, we established that although the assessed sensor fusion algorithm does leverage the repetitive nature of gait activities to improve performance, IMU-based joint angles can still achieve promising accuracy for single movement cycles, especially when reliable reference pose values in all three dimensions are available. Nevertheless, the possibility of varying performance in different subjects for non-repetitive motion patterns should be further investigated. Next steps shall involve in vivo experiments using a larger subject cohort, as well as an evaluation of the effects of soft tissue artefacts.

## Figures and Tables

**Figure 1 sensors-24-03324-f001:**
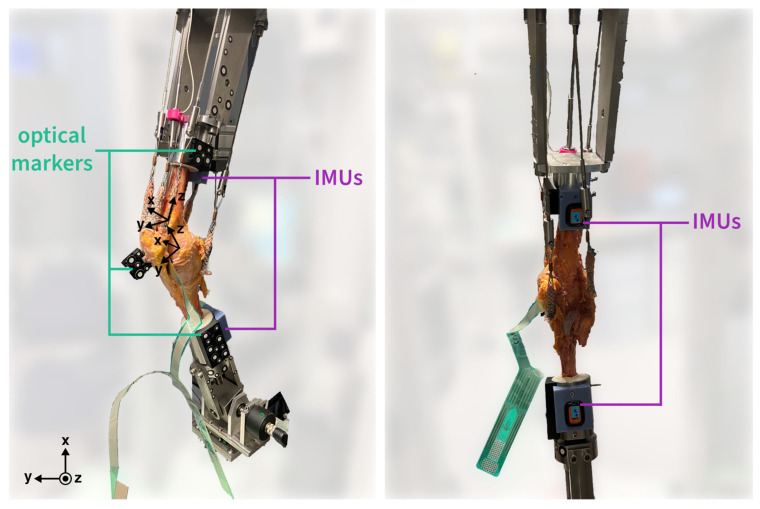
Cadaveric knee on the knee rig, side-view with optical markers (**left**) and from behind with IMU sensors (**right**).

**Figure 2 sensors-24-03324-f002:**
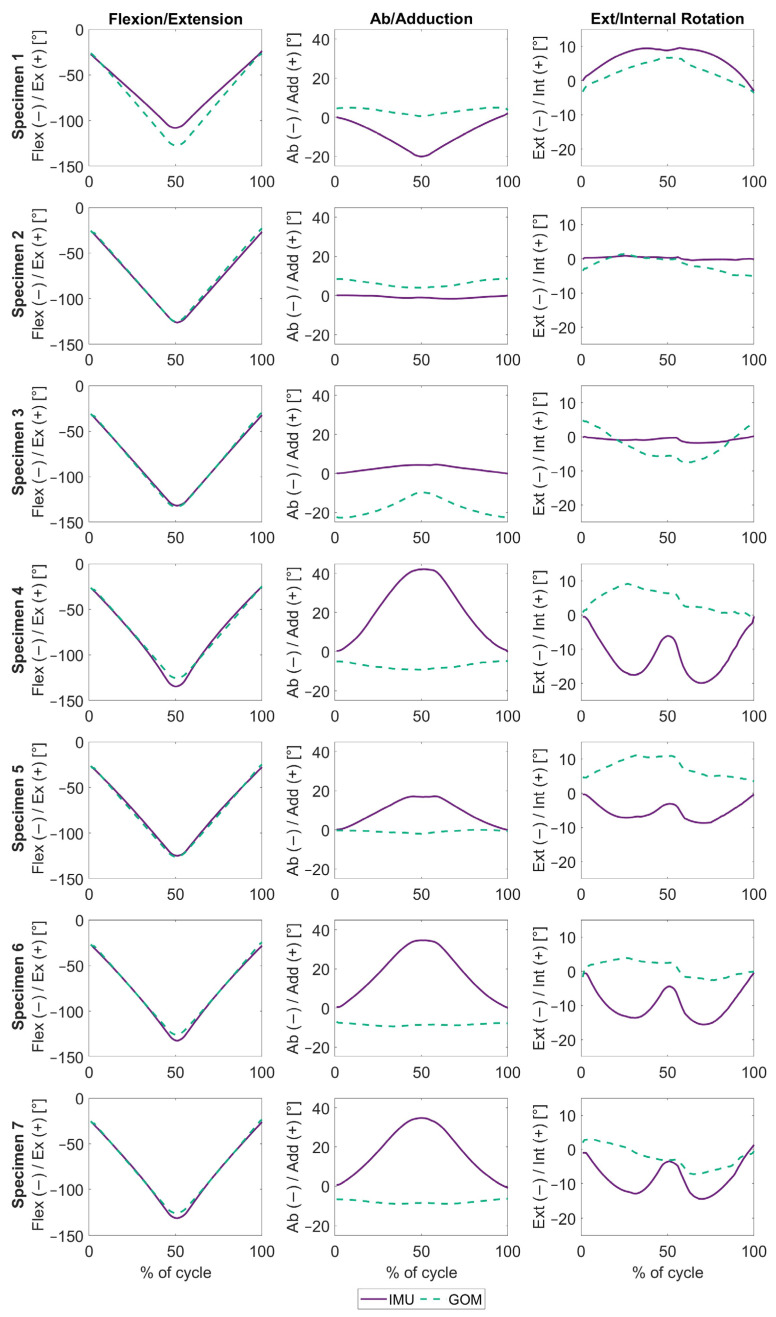
Raw knee joint angles, plotted over the entire squat movement, expressed as a percentage. The solid purple lines represent the angles estimated using inertial data. The dashed green lines represent the angles measured by the optical marker-based system. Each row represents one subject, while the columns represent flexion/extension, ab/adduction and ext/internal rotation (from **left** to **right**).

**Figure 3 sensors-24-03324-f003:**
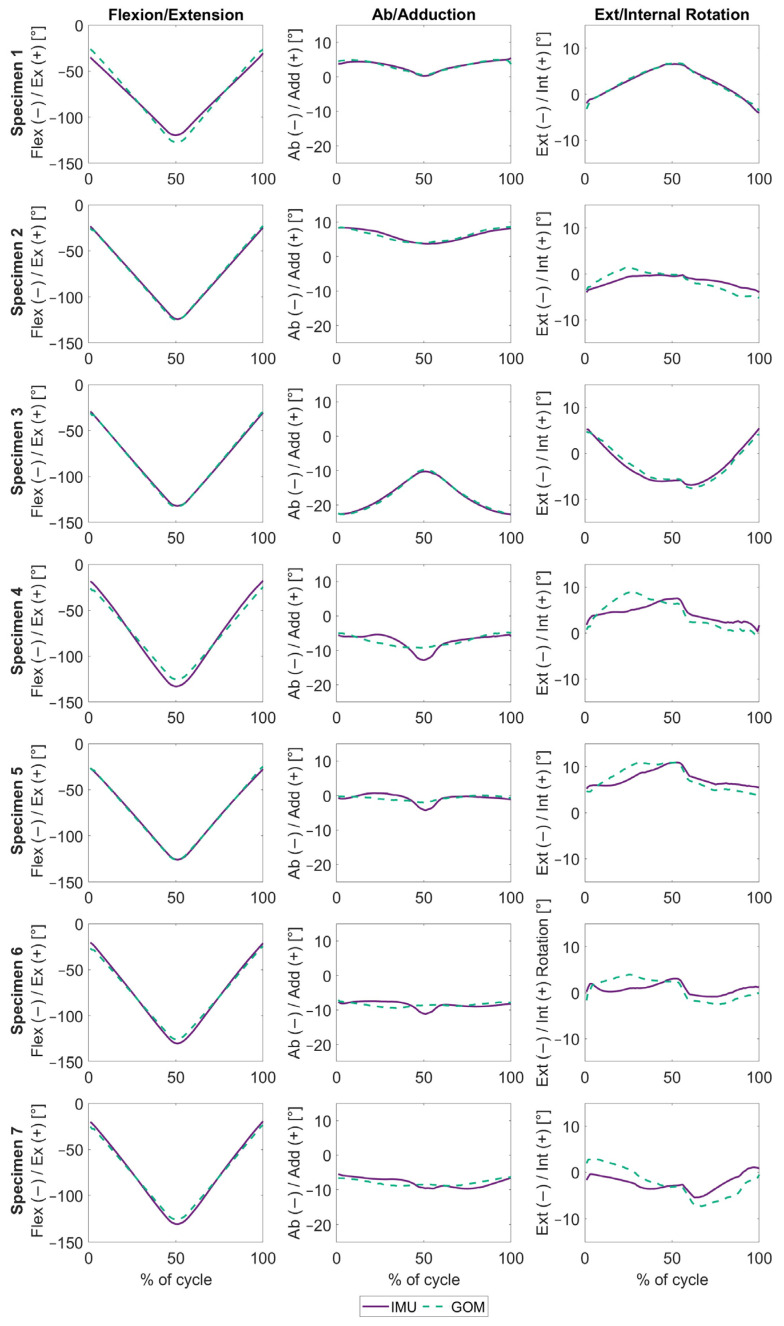
REFRAME_IMU→GOM_ knee joint angles, plotted over the entire squat movement, expressed as a percentage. The solid purple lines represent the angles estimated using inertial data. The dashed green lines represent the angles measured by the optical marker-based system. Each row represents one subject, while the columns represent flexion/extension, ab/adduction and ext/internal rotation (from **left** to **right**).

**Figure 4 sensors-24-03324-f004:**
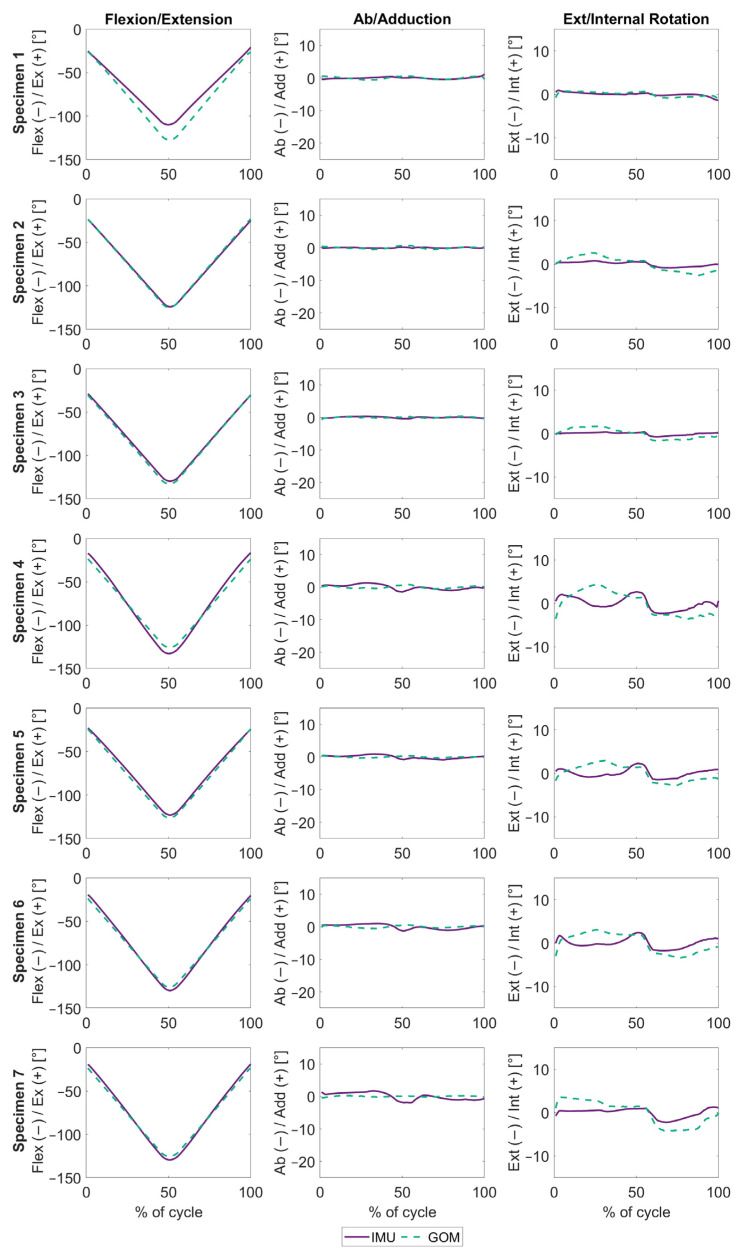
REFRAME_RMS_ knee joint angles, plotted over the entire squat movement, expressed as a percentage. The solid purple lines represent the angles estimated using inertial data. The dashed green lines represent the angles measured by the optical marker-based system. Each row represents one subject, while the columns represent flexion/extension, ab/adduction and ext/internal rotation (from **left** to **right**).

**Figure 5 sensors-24-03324-f005:**
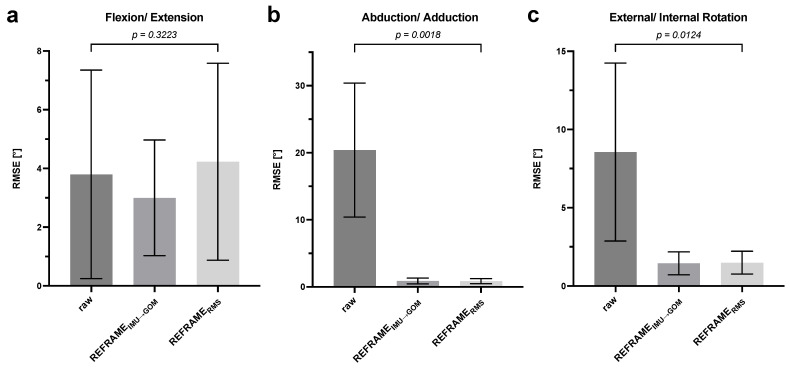
Root-mean-square error (RMSE) comparison: average root-mean-square errors ± standard deviation, before optimisation (raw), after REFRAME_IMU→GOM_ and REFRAME_RMS_: (**a**) flexion/extension, (**b**) abduction/adduction and (**c**) *external/*internal rotation. Statistical significance of differences (*p*-values) based on a paired *t*-test between raw and REFRAME_IMU→GOM_ and raw and REFRAME_RMS_.

**Table 1 sensors-24-03324-t001:** Specimen characteristics.

Specimen ID	Age (Years)	Sex	Left/Right
1	73	Male	Right
2	88	Male	Right
3	83	Female	Left
4	80	Male	Left
5	78	Female	Left
6	77	Male	Left
7	84	Female	Left
Mean	80.4		
SD	4.6		

**Table 2 sensors-24-03324-t002:** Average root-mean-square error (in degrees) for flexion/extension, ab/adduction and ext/internal rotation between IMU-based data and optical marker-based data, before and after REFRAME_IMU→GOM_ and REFRAME_RMS_.

	Flexion/Extension	Ab/Adduction	Ext/Internal Rotation
Raw	3.8 ± 3.5	20.4 ± 10.0	8.6 ± 5.7
REFRAME_IMU→GOM_	3.0 ± 2.0	0.9 ± 0.4	1.4 ± 0.7
REFRAME_RMS_	4.2 ± 3.6	0.9 ± 0.4	1.5 ± 0.7

**Table 3 sensors-24-03324-t003:** Rotational (Rot) transformations (in degrees) of the IMU-based local femoral and tibial reference frames through REFRAME_IMU→GOM_ around the x-, y- and z-axis for each specimen.

		Specimen 1	Specimen 2	Specimen 3	Specimen 4	Specimen 5	Specimen 6	Specimen 7
Femur								
Rot [°]	x	0	0	0	0	0	0	0
	y	12.5	−2.7	−6.2	30.1	14.0	23.4	23.2
	z	5.2	1.3	6.0	−1.1	−2.1	0.6	4.1
Tibia								
Rot [°]	x	−8.6	2.1	0.9	4.4	−0.8	4.0	3.2
	y	10.9	5.3	14.2	34.5	14.4	29.7	26.1
	z	9.6	−3.8	−3.2	7.4	−1.1	8.4	13.2

**Table 4 sensors-24-03324-t004:** Rotational (Rot) transformations (in degrees) of the IMU-based local femoral and tibial reference frames through REFRAME_RMS_ around the x-, y- and z-axis for each specimen.

		Specimen 1	Specimen 2	Specimen 3	Specimen 4	Specimen 5	Specimen 6	Specimen 7
Femur								
Rot [°]	x	0	0	0	0	0	0	0
	y	16.8	0.9	3.0	26.8	12.4	22.7	22.7
	z	−0.6	0.2	0.1	0.9	1.0	1.1	1.5
Tibia								
Rot [°]	x	1.0	0.0	0.0	5.5	1.0	3.6	3.1
	y	15.0	0.9	2.7	25.0	10.9	20.4	20.0
	z	7.4	0.6	1.7	8.9	5.2	8.7	10.0

**Table 5 sensors-24-03324-t005:** Rotational (Rot) transformations (in degrees) of the GOM-based local femoral and tibial reference frames through REFRAME_RMS_ around the x-, y- and z-axis for each specimen.

		Specimen 1	Specimen 2	Specimen 3	Specimen 4	Specimen 5	Specimen 6	Specimen 7
Femur								
Rot [°]	x	0	0	0	0	0	0	0
	y	3.8	3.6	9.1	−3.2	−1.7	−0.7	−0.8
	z	−4.8	−0.8	−5.1	1.1	2.9	−0.3	−3.0
Tibia								
Rot [°]	x	−0.2	0.1	−1.1	0.7	0.4	0.2	−0.2
	y	1.5	−4.7	−11.9	−8.8	−3.5	−8.4	−5.7
	z	−0.2	4.3	4.4	1.9	6.6	0.2	−3.5

## Data Availability

The raw data supporting the conclusions of this article will be made available by the authors upon reasonable request.

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
