# Peer review of "Validation of Inertial-Measurement-Unit-Based Ex Vivo Knee Kinematics during a Loaded Squat before and after Reference-Frame-Orientation Optimisation"

_sensors, 2024, doi:10.3390/s24113324_

Round 1
Reviewer 1 Report
Comments and Suggestions for Authors
The authors have established a need for this study. Background sufficient, explicit purpose, methods mostly complete, results and discussion well-presented, and conclusion relates to primary purpose and most important results. I have a few minor comments to address. See pdf

Comments on the Quality of English Language
No major issues.
Reviewer 2 Report
Comments and Suggestions for Authors
Dear Authors,
the paper is interesting and well written. I enclose the .pdf file with my minor comments inside.
Best regards

Comments on the Quality of English Language
Minor editing of English language required
Reviewer 3 Report
Comments and Suggestions for Authors
This article presents a comparative analysis of joint motion capture effects between an inertial measurement unit and an optical motion capture system, based on knee joint cadaver specimens in specific environments. The study considers the significant influence of knee ligaments on joint motion. Test results demonstrate that local coordinate optimization significantly impacts the differences in results between different motion capture systems, with implications for further enhancing the accuracy of inertial dynamic capture systems.
In order to better improve the manuscript, the following suggestions are listed for reference:
1.The direction of applying a constant 20 N muscle force described is not given in the text. Please add reference information.
2.The symbol REFRAMEIMU-GOM should explain its subscript meaning when it first appears.
3.Please further analyze and explain the reason for the significant difference in IMU and GOM lines between the original knee joint angles of ex/internal rotation in Fig 1.
4.“Additionally, frame transformations consisting of rotations around the femoral x-axis during optimisation were restricted to prevent non-physiological frame orientations.” The limitations of the direction of the frame require further clarification.
5.“Notably, the magnitude of errors affecting abduction/adduction angles were seemingly associated with joint flexion, possibly indicating the presence of crosstalk artefact.” The relationship between crosstalk artefact and error should be further elaborated. Does it indicate that there may be problems with optimization based on optical measurements?
6. “The reason for the significant difference between the IMU and GOM lines of the raw knee joint angle ab/adduction is due to crosstalk artefacts.” Could the author further explain the data that affects the raw knee joint angle ab/adduction.
7.The content of Fig 5b is difficult to read and needs to be redrawn.
8. The manuscript takes into account the influence of soft tissues such as ligaments and uses a segment model with articular ligaments in cadaveric trials, but does not further discuss the influence of ligaments on motion capture. Is it the role of ligaments that causes complex motion patterns of the knee joint, resulting in measurement errors?
